# Lymph Node Metastases Identified at the Post-Ablation 131I SPECT/CT Scan Is a Prognostic Factor of Intermediate-Risk Papillary Thyroid Cancer

**DOI:** 10.3390/diagnostics12051254

**Published:** 2022-05-18

**Authors:** Xi Jia, Yuanbo Wang, Lulu Yang, Kun Fan, Runyi Tao, Hui Liu, Xiaobao Yao, Aimin Yang, Guangjian Zhang, Rui Gao

**Affiliations:** 1Department of Nuclear Medicine, The First Affiliated Hospital of Xi’an Jiaotong University, Xi’an 710061, China; mydjx@126.com (X.J.); mars8727@xjtufh.edu.cn (Y.W.); 004125@xjtufh.edu.cn (L.Y.); yangaimin@mail.xjtu.edu.cn (A.Y.); 2Department of Thoracic Surgery, The First Affiliated Hospital of Xi’an Jiaotong University, Xi’an 710061, China; fankun1994@stu.xjtu.edu.cn (K.F.); try2083@stu.xjtu.edu.cn (R.T.); michael8039@163.com (G.Z.); 3Department of Biobank, The First Affiliated Hospital of Xi’an Jiaotong University, Xi’an 710061, China; liuh2018@xjtu.edu.cn; 4Department of Otorhinolaryngology, The First Affiliated Hospital of Xi’an Jiaotong University, Xi’an 710061, China; xiaobao4163@sina.com

**Keywords:** radionuclide imaging, persistent, recurrence, intermediate risk, papillary thyroid cancer

## Abstract

The intermediate-risk category of papillary thyroid cancer (PTC) comprises heterogeneous patients within a wide range of stages and varied care management. Understanding the risk features of recurrence observed after the initial therapy should be emphasized. We aimed to evaluate the prognostic significance of radioactive iodine-avid lymph nodes observed during the initial treatment of patients with PTC that were considered to be at intermediate risk. Data on patients with intermediate-risk PTC treated from 2012 to 2018 were retrospectively reviewed. Post-therapeutic SPECT/CT (Rx SPECT/CT) was evaluated in the enrolled patients. The clinical, pathologic, and incidence of radioiodine-avid (RAI-avid) lymph node metastasis (mLN) on Rx SPECT/CT were reviewed, and risk factors related to recurrent disease were analyzed. After a median follow-up of 37.26 (30.90, 46.33) months, structural persistent/recurrent disease was detected in 9.81% (36/367) of patients with intermediate-risk tumors. The incidence of recurrence was higher in patients who demonstrated RAI-avid mLN after the initial therapy than in those who did not (*p* < 0.001). In a multivariate Cox proportional hazard regression analysis, RAI-avid mLN appeared to be a robust risk factor for recurrent disease after the initial therapy (HR: 8.967, 95% CI: 3.433–23.421, *p* = 0.000). RAI-avid mLN is a significant risk factor for recurrent intermediate-risk PTC after the initial treatment.

## 1. Introduction

The incidence of papillary thyroid cancer (PTC) has rapidly increased worldwide in recent years, with the majority of tumors being low-to-intermediate risk [1]. The intermediate-risk category, defined by the American Thyroid Association (ATA), comprises heterogeneous patients who have malignant tumors in widely ranging stages (from T1N0 to T3N1b) [2,3]. As expected, different degrees of biological aggressiveness and various combinations of risk factors resulted in markedly varied care management for these patients [4,5]. 

All management aspects of these patients remained controversial, expanding from the initial tumor treatment to the follow-up strategy [5,6]. Perhaps more than in any other domain, studies on optimal long-term surveillance should be conducted [6]. Recently, an individualized risk assessment based on initial therapy responses was proposed to tailor the length and interval of patient surveillance [7]. Consequently, risk features detected after the initial therapy for intermediate-risk PTC, such as tumor size, multifocality, extrathyroidal extension (ETE), metastatic lymph node ratio of distant metastasis, radioiodine (RAI) avidity, and genetic mutation should be emphasized [8,9,10]. 

Most recurrences were of a loco-regional nodal disease, which was supposedly derived mainly from malignant cells that survived radioactive iodine ablation [11,12]. Therefore, metastases detected on post-therapeutic SPECT/CT (Rx SPECT/CT) could harbor lesions cause persistent or recurrent disease [8,10,11,12]. The heterogeneity of radiosensitivity in metastatic lesions, i.e., the existence of less differentiated tumor cells in RAI-avid metastatic lymph nodes (mLN), might be the origin of disease recurrence [9,13,14]. 

However, the specific value of RAI-avid mLN detected after the initial RAI as a predictor of the persistent or recurrent structural disease remains unclear to date [15,16]. Therefore, this study was performed to evaluate the prognostic value of the existence of RAI-avid mLN on Rx SPECT/CT in the individualized clinical management of intermediate-risk tumors.

## 2. Materials and Methods

### 2.1. Participants

A single-center, retrospective cohort study was conducted involving patients admitted for RAI with intermediate differentiated thyroid cancer (DTC) at the First Affiliated Hospital of Xi’an Jiaotong University between 2012 and 2018. Patients with intermediate-risk tumors according to TNM staging (eight edition) and ATA risk categories were selected [17]. Patients with tumor size of >4 cm or any size with a minimal extrathyroidal extension and without evidence of cervical lymph node metastases, as well as those with tumors of ≤4 cm who had metastatic cervical lymph nodes, were selected (T1-2N1, T3N1, or T3N0). Patients aged <17 years at the time of surgery, with poorly differentiated thyroid cancers, T4, distant metastasis detected during the primary treatment, as well as those with more than one primary cancer were excluded from the study [18]. 

All included patients underwent total thyroidectomy with or without prophylactic central neck dissection or lateral neck dissection. In one week prior to radioiodine ablation, Serum TSH, sTg, and anti-Tg antibody levels were measured by radioimmunoassay (GASK-PR, CIS-Bio International, subsidiary of Schering S.A., Gif-sur- Yvette, France). US examinations were performed by experienced radiologists for all patients before surgery using a 10–12 MHz linear transducer. Suspicious US characteristics were hyperechogenicity, cystic changes, calcification, abnormal vascularity, heterogeneous echogenicity, a round shape (longitudinal/transverse diameter ratio < 1.5), and a loss of echogenic hilum. The locations (levels II-VI) of all cervical LNs were recorded based on the guidelines of the American Joint Committee on Cancer Classification. When the results of the examiners were discordant, an agreement was reached by a joint review of the data. Gross extrathyroidal extension (gETE) was defined according to previous descriptions [19] as gross tumor invasion identified at the time of surgery and confirmed by histopathologic review. Minor extrathyroidal extension (mETE) was defined as tumor invasion beyond the thyroid capsule identified at the time of pathologic examination. Subsequent RAI remnant ablation was also conducted based on previously reported protocol [20,21]. Depending on the pathological result and the preablation examinations, administered doses of 131I ranged from 2.96 to 5.55 GBq (80–150 mCi). At the time of 131I administrations, serum thyroid-stimulating hormone levels were >30 mIU/L in all patients. Post-RAI ablation SPECT/CT was performed 5–10 days post-treatment. Rx SPECT/CT was separately analyzed by two experienced nuclear medicine physicians. A third nuclear medicine physician was consulted if there was a disagreement between these two investigators. 

### 2.2. Follow-Up Strategy

After the primary treatment, all patients received L-thyroxine (L-T4) at TSH-suppressive doses and were periodically followed up, undergoing physical examinations, serum Tg measurements, and neck ultrasonography every 3–12 months. Diagnostic WBS and stimulated Tg after L-T4 withdrawal were performed in all patients at least once. Patients clinically suspected of local recurrence or distant metastasis were assessed using CT scans, diagnostic SPECT/CT, 18F-fluorodeoxyglucose positron emission tomography, and/or fine-needle aspiration (FNA)/surgery. The clinical, pathologic, and incidence of RAI-avid mLN on Rx SPECT/CT were reviewed, and risk factors related to persistent/recurrent disease were analyzed [22]. Information on the last follow-up interval was also collected. All subjects gave written informed consent in accordance with the Declaration of Helsinki. The protocol was approved by the Ethics Committee of the First Affiliated Hospital of Xi’an Jiaotong University. 

### 2.3. Structural Persistent/Recurrent Disease

The outcome of the current study was structural persistent/recurrent disease, which was defined as a pathology-proved disease or biochemically incomplete evidence (suppressed Tg ≥ 1 ng/mL, stimulated Tg ≥ 10 ng/mL or rising anti-Tg antibody levels above 100 IU/mL) with a lesion highly suspicious of being recurrent on two serial imaging studies, as mentioned in previous studies [17,18]. The other patients were presumed to be disease-free. Recurrence-free survival was defined as the time interval (months) between the initial ablation date and the most recent follow-up date for patients without structural persistent/recurrent disease [4,7,17,18]. 

### 2.4. Statistical Analysis

Continuous data are presented as medians and ranges or means and standard deviations, as appropriate for each variable. Chi-square tests were performed for categorical comparisons, and the Mann–Whitney U test for continuous variables. Recurrence-free survival was estimated using the Kaplan–Meier method. Factors associated with the risk of persistent/recurrent disease were analyzed using Cox proportional hazard regression analysis. The following characteristics were included: age, gender, tumor diameter, multifocality, capsular invasion, extrathyroidal extension, sTg before ablation, metastatic ratio of LN resected, and post-RAI ablation SPECT/CT avidity [15,19]. Hazard ratios (HRs) and confidence intervals (CIs) were calculated in the model. All analyses were performed using SPSS version 23 (SPSS, IBM Corp., Armonk, NY, USA).

## 3. Results

### 3.1. Patient Characteristics

We screened 976 patients who underwent surgery and were diagnosed as differentiated thyroid cancer by pathology. A high proportion of low risk patients who were not routinely recommended RAI therapy by ATA as well as a cohort of lost follow-up patients were excluded. In total, 379 PTC patients with intermediate-risk tumors prepared for RAI were enrolled; 12 patients who showed distant metastasis on initial Rx SPECT/CT (7 lung metastasis, 5 bone metastases) were excluded for further analysis as they progressed into a high-risk disease. Finally, 367 patients with intermediate-risk tumors were enrolled in this study. The mean age was 44.76 ± 12.47 years, consisting of 264 women. Most tumors manifested mETE (83.38%), and 17.71% demonstrated gETE. Of the patients, 28.61% (105/367) exhibited metastatic LNs ≥ 5. The median sTg before ablation was 0.00 (0.00, 8.51) ng/mL.

The median administered RAI dose was 4.44 (3.70,4.81) GBq, and RAI-avid mLN was found in 24.25% (89/367) of patients on Rx SPECT/CT images (RAI-positive group). Meanwhile, no RAI-avid lesion was detected on Rx SPECT/CT in 75.75% (278/367) of patients (RAI-negative group). The clinical characteristics of these patients are summarized in Table 1. The gender, number of LNs positive, and preablation sTg of patients who showed RAI-avid mLN were significantly different from those of patients who did not show RAI-avid lesions in Rx SPECT/CT (*p* < 0.05, Table 1). 

### 3.2. Characteristics of Patients with Structural Persistent/Recurrent Disease

After a median follow-up of 37.26 (30.90,46.33) months, structural persistent/recurrent disease was detected in 9.81% (36/367) of patients (Table 2). All these patients were classified as intermediate risk based on the ATA risk stratification after the initial treatment. Structural diseases were detected using neck ultrasound (US), 131I scan, CT, and/or PET/CT. Most cases (n = 17) demonstrated a recurrent disease with a BIR (biochemical incomplete response) to initial therapy, while the remaining 19 cases were confirmed to have structural persistent/progressed disease, including 2 (4.35%) patients who developed bone and/or lung metastasis during follow-up. The majority of patients were a locoregional disease; one of them died seven years from the initial diagnosis. 

For patients who exhibited structural persistent/recurrent disease, additional treatments were administered, including 38 additional radioiodine treatments. Subsequent surgery was performed in 16 cases that demonstrated possible locoregional disease, which was confirmed by the pathology. One of them confirmed poorly differentiated thyroid cancer mLN. Excellent response (ER) was demonstrated in 11 (68.75%) out of the 16 cases. BIR and SIR (structural incomplete response) were 1 (6.25%) and 4 (25.00%) out of the 16 cases, respectively. 

### 3.3. Risk Factors for Structural Recurrence in the Intermediate Risk PTC

The Kaplan–Meier test was performed to evaluate prognostic factors including age, gender, tumor diameter, multifocality, capsular infiltration, extra thyroidal extension, LNs metastasis, Tg, TGAb, US before ablation, and WBS results. Patients with RAI-avid mLN had a poor prognosis as compared with those with non-RAI-avid lesion (*p* = 0.000). The log-rank test data are listed in Table 3. The median survival time of WBS results was 56.17 months for the RAI-positive and 75.27 months for the RAI-negative.

According to univariate regression analysis, gETE, sTg, US before ablation and the RAI-positive group significantly affected recurrence-free survival time (RFS) of intermediate PTC patients. The corresponding information is detailed in Table 4. RAI-positive groups, gETE, sTg, US before ablation and tumor size were independent poor prognosis factors for intermediate PTC patients by multiple regression analysis. The survival time of PTC patients in the RAI-positive group is significantly shorter than those in the RAI-negative group, indicating that intermediate PTC patients with RAI-avid mLN had a poor prognosis. The difference was statistically significant (Figure 1).

We divided the patients into two subgroups according to the results of sTg in the WBS positive cohort. Kaplan–Meier analyses were performed to compare RFS between the groups. A significantly shorter RFS was found in patients who demonstrated sTg positive (38.507 months vs. 53.622 months, Log rank test *p* = 0.002, Figure 2) in RAI-avid mLN. Cox proportional hazard multivariate regression analysis for variables associated with structural persistent/recurrent disease was performed (Table 5). The risk of structural persistence/recurrence increased with the tumor size (HR: 2.468, 95% CI: 1.065–5.719, *p* = 0.035), as well as the gETE (HR: 7.453, 95% CI: 1.650–33.659, *p* = 0.009).

## 4. Discussion

Intermediate risk includes a wide range of stages in patients with PTC; thus, an individualized post-initial therapy risk stratification is needed for the optimal follow-up strategy of patients [7,8,9]. The Rx SPECT/CT was usually used to complete postoperative staging, and information obtained may modify the treatment plan and long-term surveillance [10,11]. The avid-mLNs presented an advanced situation of PTC. Its prognostic value for individual patients remains unclear. This study therefore aimed to investigate the exact predictive value of radioiodine-positive mLN on the occurrence of persistent/recurrent disease.

The novelty of our study was to retrospectively evaluate the prognostic significance of Rx SPECT/CT-positive LN in PTC patients who were considered at intermediate risk (from T1N0 to T3N1b) [2,3]. After a median follow-up of 37.26 months, structural persistent and/or recurrent disease was detected in 9.81% (36/367) of patients. When patients were classified based on Rx SPECT/CT findings, structural persistent/recurrent disease was significantly higher in the RAI-avid mLN group (22/89, 24.72%). Only 5.04% (14/278) of patients in the RAI-negative group had structural persistent/recurrent disease. Our data indicate that short- to medium-term outcomes (no more than 5 years of the median follow-up period) are less favorable in patients who demonstrated RAI-avid mLN in comparison to patients who did not. RAI-avid nodal metastasis revealed at initial therapy would lead to a significantly shorter RFS in PTC (*p* < 0.001, HR, 8.967). Our data showed that only 5.04% (14/278) of patients without RAI-avid lesions at radioablation developed persistent/recurrent disease during follow-up. This is in line with the report from Schmidt D. et al., in which patients without Rx SPECT/CT mLN at radioablation had no 131I radioiodine-positive LN during the 5-month follow-up. [10] Negative Rx SPECT/CT may therefore be used to indicate a better prognosis in PTC with an intermediate tumor [10,23]. Large-scale studies with longer follow-up are needed to address how SPECT/CT performed at radioablation can be considered as one of several predictive parameters that can best be integrated into the follow-up procedure for PTC. 

It is of note that newly identified RAI-avid lesions were demonstrated in 19 patients in the RAI-avid mLNs group, which is significantly higher than that of the non-RAI-avid-lesion group (19/89 vs. 2/278, *p* = 0.000). The newly identified RAI-avid lesions might originate from the less differentiated tumor cells, such as cancer stem cells, that survived initial RAI. They progressed to demonstrate radioiodine concentration ability during the follow-up period [24]. Moreover, we should be aware that the existence of less differentiated tumor cells may lead to a higher possibility of iodine-refractory disease. After a systemic treatment, 7 out of the 19 patients in the RAI-avid mLNs group still showed a structural incomplete response (median follow-up: 53 months). One of the recurrence/persistent origins was the iodine-refractory thyroid cancer cell that coexisted with the RAI-avid mLNs [25]. 

We further investigated the prognostic value of sTg in the Rx SPECT/CT positive group. The result showed that the patients with RAI-avid but sTg-negative nodes have a better prognosis than those who revealed RAI-avid and sTg-positive nodes (*p* = 0.017). The 131I-detectable mLN with negative sTg had a significantly better chance to be eliminated by initial RAI. While RAI-avid nodes with elevated sTg had a higher chance of harboring tumor cells demonstrating heterogeneous radiosensitivity, this might be an origin for future recurrence, as mentioned above [26]. 

In this present study, the RFS of intermediate-risk patients was 9.81%, which is relatively higher than in previous studies [27,28]. Various risk classification, follow-up periods, as well as the different extent of surgical dissection might attribute to the difference. Residual nodal metastasis identified in Rx SPECT/CT was mainly considered due to the incomplete removal of metastatic lymph nodes during the initial surgery. Thus, the incidence of Rx SPECT/CT mLN was inherently limited according to the extent of surgical dissection [7,29,30]. Median resected LNs in T2 and T3 tumors in this study were 9 (range 1–112), which is in the range of values reported in the literature [19]. However, according to Robinson et al.’s report, at least 9 LNs were suggested to be resected for patients with T2 tumors, and 18 LNs for patients with T3 disease to rule out the possible incomplete removal of mLNs [31]. The number of resected LNs in our study may not sufficiently rule out occult nodal metastatic disease. This might be the source of RAI-avid mLN observed on the preablation US or Rx SPECT/CT, and metastatic tumors that survived RAI are the most common etiology for persistent/recurrent disease [8,10,11,12]. Even more, this could be an important reason for the higher recurrence rate in this present study. This also could be an important reason for the relatively high recurrence rate in short- to medium-term outcomes, although it was consistent with previous studies [9,13,14,32]. The most obvious difference of RFS came from the first two years; these early recurrences are associated with high mortalities. Rigorous follow-up and more active treatments are needed to reduce the possibility of recurrence in cases that showed RAI-avid mLN [33].

This study had limitations due to its retrospective design nature and the fact that patients from a single tertiary referral center were selected. Only patients with intermediate-risk PTC who were referred for RAI treatment were included in this study, which introduced an inherent selection bias. The intermediate-risk was evaluated by post-pathological evidence, while patients with low recurrence risk evaluated postoperatively might progress to intermediate risk as RAI-positive findings. The sample size and the follow-up time are other major limitations. Finally, 367 patients met the included criterion. Although the sample size is large enough regarding the calculation results, a large-scale cohort is necessary for validation in the future. 

## 5. Conclusions

The incidence of RAI-avid mLN on Rx SPECT/CT was relatively high in intermediate-risk PTC. RAI-avid mLN has a high predictive value regarding structural persistent/recurrent disease after the initial therapy. Based on the results of this study, a more active follow-up strategy would be suggested for patients with PTC who have RAI-avid mLN, especially in the first two years after initial therapy.

## Figures and Tables

**Figure 1 diagnostics-12-01254-f001:**
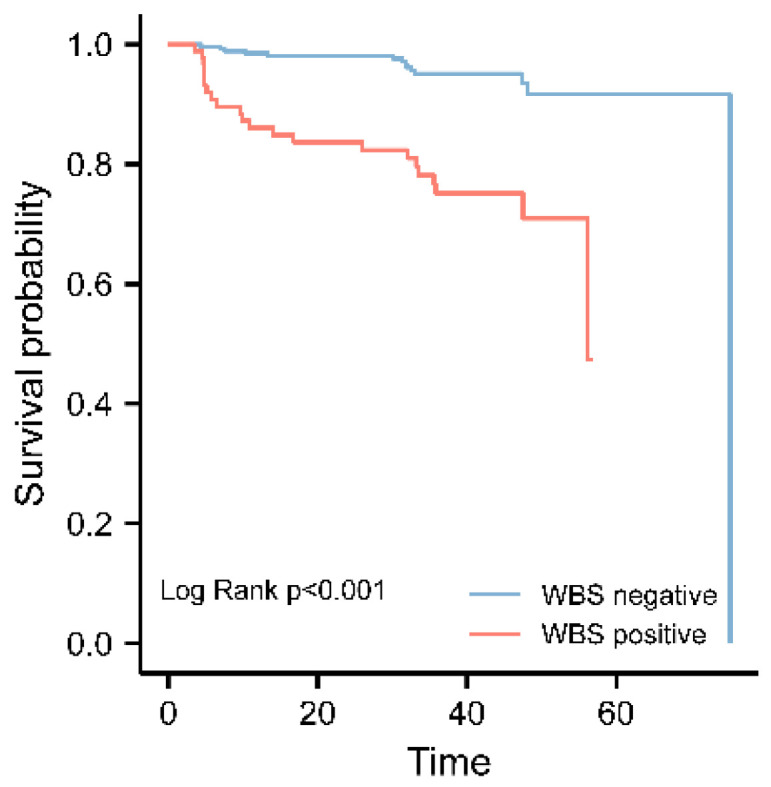
Prognostic value of RAI-avid lymph node metastasis revealed on Rx SPECT/CT.

**Figure 2 diagnostics-12-01254-f002:**
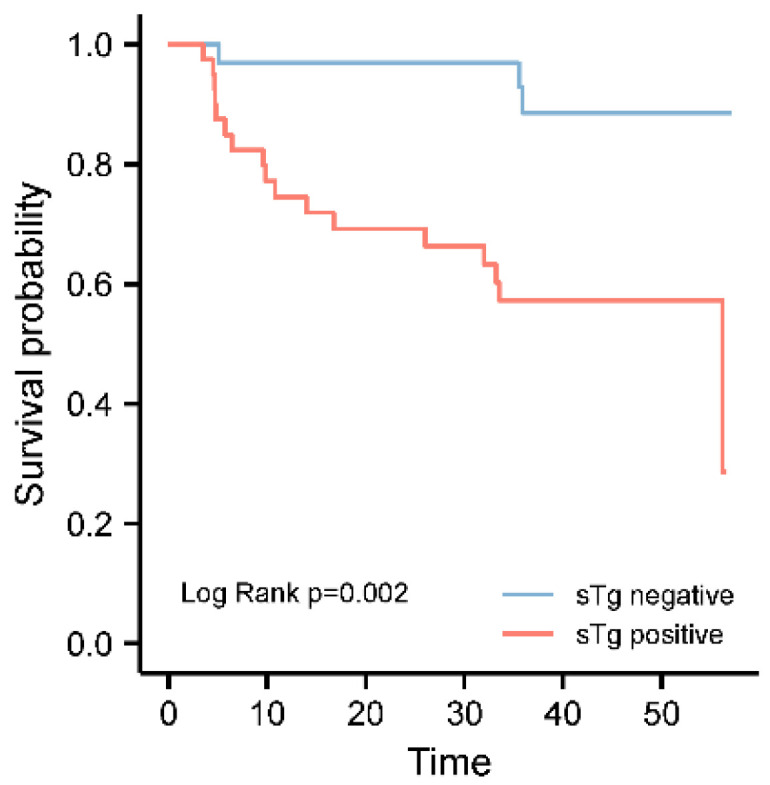
Prognostic value of US suspected mLN in RAI-positive group.

**Table 1 diagnostics-12-01254-t001:** Patient characteristics.

		All Cases n = 367	RAI-Avid mLN + n = 89	RAI-Avid mLN- n = 278	*p*
Age, years			44.22 ± 12.14	44.93 ± 12.59	0.642
Gender	Male	103	36	67	0.003 *
	Female	264	53	211	
Tumor size, cm			1.70 (1.50, 2.00)	1.80 (1.30, 2.50)	0.574
Multifocality	Absent	203	46	157	0.429
	Present	164	43	121	
gETE	Absent	302	70	232	0.302
	Present	65	19	46	
mETE	Absent	61	14	47	0.795
	Present	306	75	231	
Subtype	Aggressive	4	1	3	0.809
	Non-aggressive	362	88	274	
	Others	1	0	1^#^	
LNs metastasis	≥5	113	31	82	0.343
	<5	254	58	196	
US before ablation	Positive	33	12	21	0.089
	Negative	334	77	257	
Number of LNs positive			3.00 (1.00, 6.00)	2.00 (0.00, 6.00)	0.047 *
Number of LNs resected			7.00 (3.00, 15.00)	8.00 (3.00, 19.25)	0.840
sTg before RAI, ng/mL		0.00 (0.00, 8.51)	6.40 (0.00, 22.90)	0.00 (0.00, 3.22)	0.000 *
TGAb (+)			2	10	0.533
Follow-up period			39.00 (31.45, 47.87)	37.03 (30.85, 46.00)	0.447

RAI-positive group: positive RAI-avid lymph node metastasis revealed on Rx SPECT/CT; RAI-positive group: no RAI-avid lesion revealed on Rx SPECT/CT; gETE: gross extrathyroidal extension; mETE: minor extrathyroidal extension; LN: lymph node; sTg: TSH stimulated thyroglobulin level; RAI: radioactive iodine ablation; TGAb (+): TGAb ≥ 100 IU/mL; follow-up period: follow-up since initial treatment. ^#^ Coexistence of primary squamous cell carcinoma of thyroid with PTC; * *p* < 0.05.

**Table 2 diagnostics-12-01254-t002:** Clinicopathologic characteristics of patients with structural recurrences (n = 36).

Recurrent/Persistent Cases		RAI(+) n = 22	RAI(-) n = 14	*p* Value
Age, y		43.14 ± 13.00	48.86 ± 15.03	0.270
Gender	Male	11	3	0.086
	Female	11	11	
Tumor size, cm		2.00 (1.75, 3.00)	1.75 (1.20, 2.63)	0.215
Multifocality	Absent	10	8	0.494
	Present	12	6	
gETE	Absent	14	9	0.968
	Present	8	5	
mETE	Absent	3	2	0.956
	Present	19	12	
Subsequent surgery		5	11	0.010 *
LNs metastasis	≥5	8	7	0.418
	<5	14	7	
US before ablation	Positive	6	5	0.592
	Negative	16	9	
sTg at RAI, ng/mL		30.51 (10.00, 113.00)	15.82 (1.60, 73.97)	0.548
TGAb (+)	Positive	1	2	0.303
	Negative	21	12	
Follow-up period		32.60 (14.86, 46.46)	33.10 (31.61, 45.03)	0.395

sTg: stimulated Tg before radioiodine ablation; RAI: radioactive iodine ablation; RAI-positive group: positive RAI-avid lymph node metastasis revealed on Rx SPECT/CT; RAI-negative group: no RAI-avid lesion revealed on Rx SPECT/CT; gETE: gross extrathyroidal extension; mETE: minor extrathyroidal extension; TGAb (+): TGAb ≥ 100 IU/mL; * *p* < 0.05.

**Table 3 diagnostics-12-01254-t003:** Log-rank analysis of clinical parameters.

Prognostic Parameters	Log-Rank	*p* Value
Age	1.311	0.252
Gender	1.834	0.176
Tumor diameter (>4cm)	48.765	0.000 *
Multifocality	0.475	0.490
gETE	7.699	0.006 *
mETE	0.765	0.382
LNs metastasis	3.102	0.078
sTg (>10ng/mL)	38.537	0.000 *
TGAb+	3.027	0.082
US before ablation	18.127	0.000 *
RAI-positive	29.871	0.000 *

gETE: gross extrathyroidal extension; mETE: minor extrathyroidal extension; LNs metastasis: lymph node metastasis ≥5; sTg: stimulated Tg before radioiodine ablation; TGAb (+): TGAb ≥ 100 IU/mL; RAI-positive group: positive RAI-avid lymph node metastasis revealed on Rx SPECT/CT. * *p* < 0.05.

**Table 4 diagnostics-12-01254-t004:** Clinical factors associated with the RFS by Cox regression analysis.

Parameters	Univariate Analysis			Multivariate Analysis		
	HR	95.0% CI	*p*-Value	HR	95.0% CI	*p*-Value
Age	1.004	0.978–1.030	0.784	0.993	0.956–1.031	0.700
Gender	1.820	0.925–3.581	0.083	0.780	0.286–2.123	0.626
Tumor size	1.307	0.892–1.916	0.170	1.769	1.027–3.046	0.040*
Multifocal	1.347	0.694–2.614	0.379	0.815	0.316–2.098	0.671
LNs metastasis	1.610	0.818–3.167	0.168	1.204	0.453–3.196	0.710
gETE	3.080	1.548–6.129	0.001 *	3.578	1.347–9.508	0.011 *
mETE	0.733	0.457–3.042	1.179	1.539	0.368–6.441	0.555
LN ratio	2.865	0.936–8.763	0.065	1.883	0.384–9.235	0.435
sTg	1.007	1.004–1.009	0.000 *	1.007	1.002–1.012	0.005 *
TGAb	1.868	0.568–6.144	0.304	2.415	0.618–9.444	0.205
US before ablation	6.831	3.328–14.020	0.000 *	5.202	1.915–14.130	0.001 *
RAI-positive	6.059	3.075–11.940	0.000 *	8.967	3.433–23.421	0.000 *

sTg: stimulated Tg before radioiodine ablation; gETE: gross extrathyroidal extension; mETE: minor extrathyroidal extension; RAI-positive group: positive RAI-avid lymph node metastasis revealed on Rx SPECT/CT; LN ratio: metastatic ratio of LN resected; LNs metastasis: lymph node metastasis ≥5; TGAb: thyroglobulin antibody, * *p* < 0.05.

**Table 5 diagnostics-12-01254-t005:** Cox regression analysis of clinical parameters in RAI-positive group.

	Multivariate Analysis		*p*-Value
	HR	95.0% CI	
Age	1.001	0.951–1.054	0.963
Gender	1.858	0.502–6.883	0.354
Tumor size	2.468	1.065–5.719	0.035 *
Multifocal	0.615	0.145–2.605	0.509
Lymph node metastasis	0.296	0.061–1.445	0.296
gETE	7.453	1.650–33.659	0.009 *
mETE	0.754	0.099–5.726	0.784
LN ratio	1.353	0.119–15.340	0.807
Tg	1.013	1.002–1.024	0.017 *
TGAb	1.831	0.176–19.041	0.613
US before ablation	1.759	0.688–4.498	0.239

gETE: gross extrathyroidal extension; mETE: minor extrathyroidal extension; LN ratio: metastatic ratio of LN resected; pN1: pathological lymph node disease; TGAb: thyroglobulin antibody; * *p* < 0.05.

## Data Availability

The data presented in this study are available from the corresponding author upon reasonable request.

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
