# Peer review of "Lymph Node Metastases Identified at the Post-Ablation 131I SPECT/CT Scan Is a Prognostic Factor of Intermediate-Risk Papillary Thyroid Cancer"

_diagnostics, 2022, doi:10.3390/diagnostics12051254_

Round 1

Reviewer 1 Report

The manuscript is well written. In the method section Structural persistent/recurrent disease should be better dedined providing detailed information on the Tg cut off values to indicate biocemical incomplete response and Ab(+) definition. In the results overall sTg at the time of RAI should be reported in the text and tables. Tumor size is not clearly expressed in tables (1000 cm???). I suppose there was a mistyping. In table 1 does the variable" subsequent surgery " refers to surgery after RAI or after initial surgery before RAI? If it is after RAI it should be considered in the outcome and not in the initial characteristics of baseline population. 

Author Response

Thank you for considering the revised version of our manuscript for publication in DIAGNOSTICS. We appreciate the corrections and comments from the reviewers. We have carefully revised the manuscript per these comments. Revised text has been marked out using the “Track Changes”. The point-by-point response to the reviewers’ comments was enclosed in the response letter. 

  1. The manuscript is well written. In the method section Structural persistent/recurrent disease should be better defined providing detailed information on the Tg cut off values to indicate biochemical incomplete response and Ab(+) definition.

Response: We are grateful for the suggestion. To be more clear and in accordance with the reviewer concerns, the corresponding part were revised (Line 110-112).

  1. In the results overall sTg at the time of RAI should be reported in the text and tables.

Response: Thanks for your helpful advise. The corresponding part were revised in the text and tables (Line 139-140).

  1. Tumor size is not clearly expressed in tables (1000 cm???). I suppose there was a mistyping.

Response: Sorry for the mistake and the revision was made (Table 2).

  1. In table 1 does the variable" subsequent surgery " refers to surgery after RAI or after initial surgery before RAI? If it is after RAI it should be considered in the outcome and not in the initial characteristics of baseline population. 

Response: Thanks for the suggestion and the corresponding part was deleted in Table 1.

Reviewer 2 Report

The study by Xi Jia et al has a potential, but needs some revision before considering it for publication. My commentaries:

Line 63: DTC – What types of cancer were taken into account? Since the title and abstract sections suggest PTC
also please make sure that abbreviations are explained when they are first introduced in the manuscript

Line 65: (version 8) – did you mean „eight edition”?

Section 2, materials and methods: what was the thyroglobulin level at the time of ablation?

Line 106: „mentioned above[17,18]” – did you mean “mentioned in previous studies”?

Table 1: “Extra thyroidal extension” – did you mean “extrathyroidal extension”? Additionally, please provide data on the number of gross/minor extrathyroidal extension in this table and in table 3

Table 1: can authors provide data on subtypes of PTC? (aggressive/non aggressive)

Line 163: DTC – did you mean DTC or PTC?

Line 188: recurrence-free time (RFS) – did you mean “recurrence free survival”?

Table 4: LN ratio, pN1, TGAb   – please explain these abbreviations and make sure all the abbreviations in all the tables are explained

Line 218: DTC – again: DTC or PTC?

Line 218: RFS in DTC (p<0.001, HR, 4.948) – perhaps the title of Table 4 could be changed to indicate that the results in that table are associated with the RFS

Line 236: RAI-aivd + line 242: 131I-avid – please keep the nomenclature consistent

Line 260: DTC – what kind of patients were included into the study? DTC or PTC?

Line 273: We screened 976 patients (…) with DTC – perhaps this should also be mentioned (described) in the “Materials and Methods” section (as that section mentions only 379 PTC patients)

Line 282: The incidence of RAI-avid mLN on Rx SPECT/CT was relatively high in intermediate-risk DTC. – this fragment is inconsistent with the title of the manuscript (DTC vs PTC)

Author Response

 Thank you for considering the revised version of our manuscript for publication in DIAGNOSTICS. We appreciate the corrections and comments from the reviewers. We have carefully revised the manuscript per these comments. Revised text has been marked out using the “Track Changes”. The point-by-point response to the reviewers’ comments was enclosed in the response letter. 

  1. Line 63: DTC – What types of cancer were taken into account? Since the title and abstract sections suggest PTC also please make sure that abbreviations are explained when they are first introduced in the manuscript.

Response: Thank you for the question. Papillary thyroid cancer was taken into account and the abbreviations were explained when they are first introduced in the manuscript.(Line 63, Line 22, Line 83,85, Line 187-188)

  1. Line 65: (version 8) – did you mean “eight edition”?

Response: Yes. Thank you and we revised the corresponding parts accordingly.(Line 65)

  1. Section 2, materials and methods: what was the thyroglobulin level at the time of ablation?

Response: Thank you for figuring this out and the thyroglobulin level prior to radioiodine ablation was listed in Table 1.(Table 1, Line 73-74, 139-140)

  1. Line 106: „mentioned above[17,18]” – did you mean “mentioned in previous studies”?

Response: Sorry for the negligence and the corresponding parts were revised accordingly.(Line 113)

  1. Table 1: “Extra thyroidal extension” – did you mean “extra thyroidal extension”? Additionally, please provide data on the number of gross/minor extra thyroidal extension in this table and in table 3

Response: Thank you for providing your constructive comments. We provided the definition and the data on the number of gross/minor extra thyroidal extension in text and tables. ( Line 83-86, Table 1-5)

  1. Table 1: can authors provide data on subtypes of PTC? (aggressive/non aggressive)

 Response: Thanks for the question. We presented the data on aggressive/non aggressive subtype in Table 1. The aggressive subtype included columnar/clear/eosinophilic cell variant. The non aggressive subtype included follicular variant and classical PTC. The other included one case diagnosed with coexistence of primary squamous cell carcinoma of thyroid with PTC.

  1. Line 163: DTC – did you mean DTC or PTC?

Response: Sorry for the negligence and the corresponding parts were revised.(Line 175, 217, 235, 240, 243,286, 300, 302)

  1. Line 188: recurrence-free time (RFS) – did you mean “recurrence free survival”?

Response: Thank you for pointing this out and the corresponding parts were revised accordingly.(Line 187-188)

  1. Table 4: LN ratio, pN1, TGAb – please explain these abbreviations and make sure all the abbreviations in all the tables are explained

Response: Thank you for the reminder and the abbreviations were added in Table 1- 5.

  1. Line 218: DTC – again: DTC or PTC?

Response: Thank you for underlining this deficiency and the corresponding parts were revised. (Line 175, 217, 235, 240, 243,286, 300, 302)

  1. Line 218: RFS in DTC (p<0.001, HR, 4.948) – perhaps the title of Table 4 could be changed to indicate that the results in that table are associated with the RFS

 Response: Thanks for the great suggestion and the title of Table 4 was revised.

  1. Line 236: RAI-aivd + line 242: 131I-avid – please keep the nomenclature consistent

Response: Sorry for the negligence and the corresponding parts were revised.(Line 255-256, 258-259)

  1. Line 260: DTC – what kind of patients were included into the study? DTC or PTC?

Response: Sorry for the negligence and the corresponding parts were revised. (Line 175, 217, 235, 240, 243,286, 300, 302)

  1. Line 273: We screened 976 patients (…) with DTC – perhaps this should also be mentioned (described) in the “Materials and Methods” section (as that section mentions only 379 PTC patients)

 Response: We have modified the corresponding parts according to the comment.  (Line 130-132, 290-293)

  1. Line 282: The incidence of RAI-avid mLN on Rx SPECT/CT was relatively high in intermediate-risk DTC. – this fragment is inconsistent with the title of the manuscript (DTC vs PTC)

Response: Sorry for the negligence and the corresponding parts were revised. (Line 175, 217, 235, 240, 243,286, 300, 302)

Round 2

Reviewer 2 Report

Most of my comments have been satisfactorily replied to, some minor commentaries still have to be resolved:

Table 1: sTg before RAI, ng/mL 0.00 (0.00,8.51) - are the values in brackets median or range?

Table 4, table 5: Perhaps it would look better to put mETE and gETE in consecutive lines (one under the other?)

Author Response

We would like to thank you for your careful reading, helpful comments, and constructive suggestions, which have significantly improved the presentation of our manuscript.

Table 1: sTg before RAI, ng/mL 0.00 (0.00,8.51) - are the values in brackets median or range?

Response: Thank you for the reminder. The data are presented as median and the corresponding part was revised in the text.

Table 4, table 5: Perhaps it would look better to put mETE and gETE in consecutive lines (one under the other?)

Response: We are grateful for the suggestion. To be more clear and in accordance with the reviewer‘s concerns, the corresponding part was revised.